# OpenReview forum: "In-context Learning in Presence of Spurious Correlations"
_TMLR — Accepted by TMLR_

### Review · Reviewer_2z2i · 2025-11-11

**Summary Of Contributions:**

**Summary**
---------------
This paper investigates the challenge of training Transformer-based in-context learners that are robust to spurious correlations. The authors demonstrate that conventional training approaches fail, leading to "in-weights learning" where the model memorizes the task instead of leveraging the context, and remains susceptible to shortcut features. To overcome this, they propose a novel training methodology that involves permuting input embedding dimensions to prevent memorization and forming ICL sequences with intermediate, group-balanced queries to simulate a distribution shift. Their experiments show that this approach can yield a specialized in-context learner for a single task that outperforms strong baselines like ERM and GroupDRO. Furthermore, by training on a diverse dataset of synthetic tasks, they successfully create a more general-purpose learner that can identify and ignore spurious features in unseen tasks. However, the work also critically highlights that this learned algorithm is brittle, failing completely when faced with spurious correlations that are significantly more severe than those seen during training, thereby outlining a key limitation in its ability to handle strong distribution shifts.

**Strengths:**
---------------
1. The experimental results on Waterbirds and Waterbirds-severe are strong.
2. The authors don't oversell their results. They dedicate significant effort to understanding and reporting on the "brittleness" of their learned algorithm, which is a crucial and sobering finding for the field.

**Weakness**
---------------
1. The proposed method, particularly the use of group-balanced queries, requires access to spurious feature labels during the training phase. This is could be a limitation, as such annotations are often costly or unavailable.
2. According to Figure 2(a), the performance gains appear to come almost entirely from permuting the input embeddings (+P), with the proposed intermediate query mechanism offering little-to-no additional benefit. The authors' argument is that the image embeddings of DINOv2 have a bias toward representing objects more than backgrounds. While this specific experiment is convincing, the paper's argument for the general utility of intermediate queries would be further strengthened by demonstrating this separation on existing real-world benchmarks with naturally strong spurious correlations, rather than relying on a single synthetically hardened one.

**Audience:**

Yes

**Audience Explanation:**

Yes. Since the advent of large language models, understanding the mechanism and limits of in-context learning has become a central research question. The findings of this paper would be of high interest to a substantial portion of TMLR's audience.

**Claims And Evidence:**

Yes

**Claims Explanation:**

The claims made in the submission are, for the most part, supported by accurate, convincing, and clear evidence. The authors do a good job of providing specific experiments to back up their arguments.

**Requested Changes:**

For weakness 2, I suggest the users to rerun the Waterbirds experiment using weaker, pre-trained image embeddings. If your hypothesis is correct, these less-robust embeddings should be more susceptible to the spurious background feature. In this setting, we would expect the performance gap between Naive + P and Proposed + P to widen considerably, thus providing strong, direct evidence for the utility of intermediate queries when the representation itself is not robust.

---

### Review · Reviewer_vLLM · 2025-11-25

**Summary Of Contributions:**

This paper studies the in-context learning of classification tasks in the setting where spurious correlations lead to poor accuracy on underrepresented subgroups. The standard approach of generating ICL instances is shown to induce in-weights learning, and a method based on intermediate query tokens is proposed as an alternative. Under this data generation procedure, and with regularization techniques such as permutation, in-context learners are shown to be robust to spurious correlations. Unlike robustness algorithms such as GroupDRO, it is observed that these in-context learners adapt to task structure and so do not generalize well out-of-distribution; training on a diverse set of ICL instances is proposed as a method to mitigate this.

Summary of strengths:
1. This paper’s empirical investigation is comprehensive and rigorous.
2. The paper has potential to bring two distinct research communities closer together.
3. The insight that ICL aggressively leverages task structure, leading to worse OOD generalization than conventional robustness algorithms, is particularly interesting and could motivate future research.

Summary of weaknesses:
1. There are a few methodological decisions that would benefit from further justification (see Requested Changes).
2. Inclusion and discussion of various references could be improved (see Requested Changes).

**Additional Comments:**

1. I believe the Akyurek et al citation has the wrong year.
2. Please explain the +P and +I notations in the figure captions and/or reference to where they are defined in the main text.
3. Is the group structure where groups 0 and 2 are majorities while groups 1 and 3 are minorities, referenced in the beginning of Section 2, only for Waterbirds or is this defined for general datasets? It also only holds for binary classification, and doesn’t make sense in situations where the spurious correlation applies to only one class (e.g., CelebA hair-color).
4. It would be nice to give the “Proposed” method a name for future papers to reference.
5. The first two paragraphs of Section 2.4 would benefit from a figure; I found it difficult to keep track of the benchmarks, baselines, and accuracies.

**Audience:**

Yes

**Audience Explanation:**

A strength of this paper is its potential to bring two distinct research communities closer together, namely in ICL/meta-learning and group robustness. Each community would likely be interested in this paper: the ICL/meta-learning community for its challenging generalization of standard ICL, observations on how ICL adapts to task structure, and application to relevant theories such as induction heads; the group robustness community for its new problem setting using existing datasets and comparison between ICL and conventional robustness algorithms. I expect future lines of research to address open problems raised by this paper, e.g., whether in-context learners can adapt to problems with unknown group structure, missing spurious annotations, or multiple correlations.

**Broader Impact Concerns:**

I have no broader impact concerns.

**Claims And Evidence:**

Yes

**Claims Explanation:**

Overall, the claims in the paper are well-substantiated. Experiments are performed on several different benchmark datasets at varying levels of granularity and difficulty; the experimental designs are detailed and clear; the analysis is comprehensive and addresses natural questions which may arise while reading the paper; the appendix handles extensions such as larger networks, group-balancing, and checking data leakage.

**Requested Changes:**

Critical for acceptance: None

Would strengthen the work:
1. It’s unclear to me why visual classification tasks were chosen as the primary benchmarks for ICL under spurious correlations given the proclivity of autoregressive Transformers to sequential language-based tasks. I would appreciate further justification on this point. Moreover, while not strictly necessary for acceptance, I believe it would improve the work to evaluate ICL on language benchmarks like CivilComments or MultiNLI.
2. While the permutation method for regularizing ICL memorization is reasonable, why were standard data augmentation techniques not used given the computer vision setting? For example, instead of computing and storing 10K rotation matrices as mentioned in the text, one could rotate (or flip, jitter, etc) the image itself and compute its DINO-ViT embedding. Also, it would be interesting to comment on whether the necessity of regularization for robustness in ICL has any relation to the necessity of regularization in GroupDRO.
3. Given the bias of DINO image embeddings towards objects instead of backgrounds, the paper proceeds with increasing the difficulty of Waterbirds by introducing Waterbirds-Severe. However, an alternative would have been to utilize a different image encoder, such as ResNet, which is known to be biased towards image backgrounds. Was this method considered or investigated?
4. The strategy to create Waterbirds-Severe, i.e., adding a constant vector to image embeddings, is interesting and clearly works. However, the more common method to increase difficulty in the spurious correlations literature is to increase the strength of the correlation, or in the most extreme case to have a complete correlation, where the model does not see, e.g., any waterbirds on land at train time. An investigation of this point, or justification for its absence, would improve the paper.
5. Inclusion and discussion of various references could be improved:
    * The impact of task diversity on OOD generalization is well-studied outside of the ICL literature. It may be worth discussion of, e.g., [1, 2, 3] and other related papers.
    * Missing citation and discussion of papers on task diversity and OOD generalization in ICL: [4, 5]
    * I would appreciate a more detailed contextualization with [6] in the “ICL for OOD generalization” subsection.

[1] Hendrycks et al. Pretrained Transformers Improve Out-of-Distribution Robustness. ACL 2020.

[2] Tripuraneni et al. On the Theory of Transfer Learning: The Importance of Task Diversity. NeurIPS 2020.

[3] Ramanjuan et al. On the Connection between Pre-training Data Diversity and Fine-tuning Robustness. NeurIPS 2023.

[4] Goddard et al. When can in-context learning generalize out of task distribution? ICML 2025.

[5] Raparthy et al. Generalization to New Sequential Decision Making Tasks with In-Context Learning. ICML 2024.

[6] Ahuja and Lopez-Paz. A Closer Look at In-Context Learning under Distribution Shifts. ArXiv 2023.

---

> ### Author Response · Authors · 2025-12-07
> **Author Response (Part I)**
>
> We thank the reviewer for their detailed review and helpful suggestions. We have incorporated most of the suggestions and updated the manuscript, presenting the changes in blue color.
>
> > It’s unclear to me why visual classification tasks were chosen as the primary benchmarks for ICL under spurious correlations given the proclivity of autoregressive Transformers to sequential language-based tasks. I would appreciate further justification on this point. Moreover, while not strictly necessary for acceptance, I believe it would improve the work to evaluate ICL on language benchmarks like CivilComments or MultiNLI.
>
> When designing the experiments, we wanted to be able to put 512 demonstrations in the context. To keep the overall context length short, we wanted a setting where each demonstration's input can be encoded with a single token. While this is possible for both images and text, we thought it is more natural for images. Nevertheless, we have added experiments on MultiNLI, where we encode textual inputs with a pretraining BERT. The results can be found in the updated manuscript.
>
> > While the permutation method for regularizing ICL memorization is reasonable, why were standard data augmentation techniques not used given the computer vision setting? For example, instead of computing and storing 10K rotation matrices as mentioned in the text, one could rotate (or flip, jitter, etc) the image itself and compute its DINO-ViT embedding. Also, it would be interesting to comment on whether the necessity of regularization for robustness in ICL has any relation to the necessity of regularization in GroupDRO.
>
> We did not experiment with standard data augmentations mainly because we saw that the network is able to to memorize the 10k rotations of the same encoding coming from the 10k precomuted rotation matrices. We thought that similarly the network would be able memorize the many views of the same image produced through data augmentation.
>
> > Given the bias of DINO image embeddings towards objects instead of backgrounds, the paper proceeds with increasing the difficulty of Waterbirds by introducing Waterbirds-Severe. However, an alternative would have been to utilize a different image encoder, such as ResNet, which is known to be biased towards image backgrounds. Was this method considered or investigated?
>
> We considered other image embeddings, but chose to proceed with the Waterbirds-Severe experiment to confirm our hypothesis that the results we saw on Waterbirds could be explained by DINOv2's bias towards objects.
> In response to Reviewer 2z2i's earlier request, we tried the "Naive + P" and "Proposed + P" approaches on Waterbirds with ResNet-50 image encodings and obtained the following results.
>
> | Method | Worst-group Test Accuracy | Overall Test Accuracy |
> | :------- | :------: | -------: |
> | Naive + P  | 54.03 ± 8.06 | 71.62 ± 12.21  |
> | Proposed + P | 58.42 ± 4.76  | 72.97 ± 9.08  |
>
> The results indicate that the proposed approach of having intermediate queries with balanced group distribution increases robustness to the spurious feature, although it is hard to claim this confidently due to the high standard deviations, which we attribute to less smooth optimization, marked with sudden loss decreases and long plateaus. This might be related to the fact that ResNet-50 embeddings are weaker than DINOv2 embeddings, and therefore it is harder for the model to associate inputs and labels in the context.
>
> > The strategy to create Waterbirds-Severe, i.e., adding a constant vector to image embeddings, is interesting and clearly works. However, the more common method to increase difficulty in the spurious correlations literature is to increase the strength of the correlation, or in the most extreme case to have a complete correlation, where the model does not see, e.g., any waterbirds on land at train time. An investigation of this point, or justification for its absence, would improve the paper.
>
> We considered increasing the strength of the correlation, but abstained for two reasons:
> 1. Increasing the correlation would likely not help for DINOv2 because the learner would still favor the object features due to the object bias.
> 2. Increasing the correlation would decrease the minority group sizes, and make the problem ill-defined at small context lengths, where there would be a high probability of having zero examples from a minority group.
>
>
> > Inclusion and discussion of various references could be improved:
> > * The impact of task diversity on OOD generalization is well-studied outside of the ICL literature.  It may be worth discussion of, e.g., [1, 2, 3] and other related papers.
> > * Missing citation and discussion of papers on task diversity and OOD generalization in ICL: [4, 5]
> > * I would appreciate a more detailed contextualization with [6] in the “ICL for OOD generalization” subsection.
>
> Thank you for the helpful references. We have added a "Task Diversity" paragraph in the Related Work section.

---

> > ### Author Response · Authors · 2025-12-07
> > **Author Response (Part II)**
> >
> > > I believe the Akyurek et al citation has the wrong year.
> >
> > Thank you, we have fixed it.
> >
> > > Is the group structure where groups 0 and 2 are majorities while groups 1 and 3 are minorities, referenced in the beginning of Section 2, only for Waterbirds or is this defined for general datasets? It also only holds for binary classification, and doesn’t make sense in situations where the spurious correlation applies to only one class (e.g., CelebA hair-color).
> >
> > This group structure is for all datasets (Section 2 and 3), besides CelebA. Note that the difference of CelebA does not affect the results, as we report worst-group accuracy, which naturally picks the single minority group accuracy.
> >
> > > Please explain the +P and +I notations in the figure captions and/or reference to where they are defined in the main text.
> > > It would be nice to give the “Proposed” method a name for future papers to reference.
> > > The first two paragraphs of Section 2.4 would benefit from a figure; I found it difficult to keep track of the benchmarks, baselines, and accuracies.
> >
> > We will consider these changes in future revisions.

---

### Review · Reviewer_vawz · 2025-12-04

**Summary Of Contributions:**

This paper studies in-context learning (ICL) in the presence of spurious input features (e.g., in image classification where background features correlate with the foreground label). It is worth noting that the notion of ICL used here differs from the standard usage in large language models: in this work, the model is explicitly trained on sequences containing in-context demonstrations, rather than being used as a frozen model.

The authors first conduct experiments on the Waterbirds classification task, using image features extracted from a pretrained vision model. They observe that standard ICL training leads to in-weights learning: the model ignores the in-context examples and instead simply learns the task in its parameters. To mitigate this, they introduce random permutations of input embedding dimensions (with a consistent permutation applied across all examples within an ICL instance). This forces the model to rely on contextual information and successfully induces in-context learning.

The second proposed modification inserts an additional "query" after each demonstration example. These intermediate queries are drawn from a distribution where spurious correlations do not hold. The attention mask is changed so that these queries do not influence the context seen by later tokens. The model is trained to correctly classify each intermediate query, thereby simulating a test-time distribution with no spurious correlations.

On the original Waterbirds dataset, this second modification does not improve performance. The authors attribute this to the strength of the spurious signal being too weak in the pretrained features. They therefore construct a modified "Waterbirds-severe" version by artificially amplifying the spurious background signal, and in this setting the proposed method shows performance gains.

**Weaknesses:**

1. Limited experimental scope. For the single task setup, the approach is evaluated on essentially a single dataset, and even within that dataset the main proposed modification only succeeds after artificially amplifying the spurious features. This makes the experimental setting feel somewhat synthetic and raises concerns about generality.

2. Dependence on spurious feature annotations. The method requires explicit labels for spurious features. Such annotations are rarely available in real-world applications, which limits the practicality of the approach.

3. Lack of intuition for why perturbing features improves performance. The core justification for random feature permutation is underdeveloped. It is unclear why simply perturbing input dimensions, which effectively makes the task harder, should lead to better performance than in-weights learning (with enough in-context examples), especially when only a single underlying task exists.

**Audience:**

Yes

**Audience Explanation:**

Yes, the topic of ICL in the presence of spurious features is of clear interest to the ML community.

**Claims And Evidence:**

Yes

**Claims Explanation:**

While the contribution is modest, the presentation is clear and the experimental results adequately support the claims made in the paper.

**Requested Changes:**

1. Include additional datasets for the single task setup to compare the naive baseline with the proposed approach. Those datasets should preferably be ones without the synthetic modifications introduced in the paper, to strengthen the empirical validation and demonstrate broader applicability.

2. Provide an explanation, even at an intuitive level, for why random feature perturbations lead to improved performance compared to using clean features (especially when there is only a single underlying task). This would help clarify why in-weights learning is suboptimal in this setting and why inducing in-context learning is beneficial.

---

> ### Author Response · Authors · 2025-12-07
> **Author Response**
>
> We thank the reviewer for their detailed review and helpful suggestions. We have incorporated most of the suggestions and updated the manuscript, presenting the changes in blue color.
>
> > Limited experimental scope. For the single task setup, the approach is evaluated on essentially a single dataset, and even within that dataset the main proposed modification only succeeds after artificially amplifying the spurious features. This makes the experimental setting feel somewhat synthetic and raises concerns about generality.
> > Include additional datasets for the single task setup to compare the naive baseline with the proposed approach. Those datasets should preferably be ones without the synthetic modifications introduced in the paper, to strengthen the empirical validation and demonstrate broader applicability.
>
> Please note that in the single task setting, in addition to Waterbirds, we had experimented also with CelebA. The updated manuscript adds another popular dataset: MultiNLI.
>
>
> > Dependence on spurious feature annotations. The method requires explicit labels for spurious features. Such annotations are rarely available in real-world applications, which limits the practicality of the approach.
>
> We agree that training-time access to spurious feature annotations can be a limitation. While obtaining such annotations is costly, the upside is that it is a one-time cost, and spurious feature annotations will not be needed at inference (except for in-context learners that expect group information as input, denoted with +G in the manuscript). Furthermore, we hope that future research will propose effective approaches for generating a large training dataset with synthetic spurious features, thus avoiding the costly annotation process. We provide one such approach (the grafting operation) in Section 3.
>
>
> > Lack of intuition for why perturbing features improves performance. The core justification for random feature permutation is underdeveloped. It is unclear why simply perturbing input dimensions, which effectively makes the task harder, should lead to better performance than in-weights learning (with enough in-context examples), especially when only a single underlying task exists.
> > Provide an explanation, even at an intuitive level, for why random feature perturbations lead to improved performance compared to using clean features (especially when there is only a single underlying task). This would help clarify why in-weights learning is suboptimal in this setting and why inducing in-context learning is beneficial.
>
> In the single-task regime, the mapping from inputs to labels is stable across all in-context learning training sequences.
> This enables in-weights learning, where the network encodes this particular mapping in its weights and can classify unseen queries from the same task without needing context examples.
> In other words, the learned algorithm does no in-context learning and implements a single task/mapping.
> To induce in-context learning, potentially enabling generalization to other tasks, one approach is to increase the number of training tasks.
> Permuting input dimensions is a simple approach of deriving many tasks from a single source task.
> When training with permuted input dimensions, every ICL sequences receives a different permutation and thus represents a different mapping from inputs to labels.
> This naturally encourages in-context learning, where the model uses context examples to infer the mapping on the fly, instead of encoding a single mapping.

---

> > ### Comment · Reviewer_vawz · 2026-02-03
> >
> > Thank you for the response. The response has mostly addressed my requested changes: adding a dataset (MultiNLI) and providing an intuitive explanation of why perturbing features improves performance.

---

### Decision · Action_Editor_szk1 · 2026-03-01

**Recommendation:** Accept as is

**Audience:**

Yes

**Audience Explanation:**

Yes, group robustness and ICL communities.

**Claims And Evidence:**

Yes

**Claims Explanation:**

This paper studies the problem of in-context learning (ICL) when spurious features are present. As in language models, it aims to produce ICL-like behavior in an image-based classification model. The empirical experiments are well-motivated and as one reviewer notes, all results, including negative results, are clearly reported.
Perhaps the main distinction of this paper is to combine two communities, ICL and group robustness. The paper brings some fresh ideas here that can be explored further.
Reviewers noted the limitation of a single dataset evaluation, which the authors extended during the rebuttal phase. Assumption of knowledge of spurious features is also a limitation shared by group robustness literature. While the empirical focus is narrow, I agree with the reviewers that the paper provides adequate evidence for its claims.